# Comprehensive Chloroplast Genomic Insights into *Amaranthus*: Resolving the Phylogenetic and Taxonomic Status of *A. powellii* and *A. bouchonii*

**DOI:** 10.3390/plants14050649

**Published:** 2025-02-20

**Authors:** Jizhe Han, Chuhang Lin, Tingting Zhu, Yonghui Liu, Jing Yan, Zhechen Qi, Xiaoling Yan

**Affiliations:** 1Zhejiang Province Key Laboratory of Plant Secondary Metabolism and Regulation, College of Life Sciences and Medicine, Zhejiang Sci-Tech University, Hangzhou 310018, China; hanjizhe@foxmail.com (J.H.); 13757176672@163.com (T.Z.); 202230902135@mails.zstu.edu.cn (Y.L.); 2Eastern China Conservation Centre for Wild Endangered Plant Resources, Shanghai Chenshan Botanical Garden, Shanghai 201602, China; lch798948@163.com (C.L.); yan.jing01@163.com (J.Y.)

**Keywords:** *Amaranthus* taxonomy, chloroplast genome, genome conservation, species identification, phylogenetic analyses, comparative genomics

## Abstract

*Amaranthus*, a genus in Amaranthaceae, is divided into three subgenera—*Amaranthus*, *Acnida*, and *Albersia*—and contains approximately 70 to 80 species. Understanding its phylogenetic relationships is essential for species classification, genetic diversity assessment, and evolutionary studies. This knowledge is vital for improving *Amaranthus* utilization in crop improvement and managing the ecological impacts of invasive weeds. In this study, we analyzed the chloroplast genomes of 27 *Amaranthus* species across all three subgenera to characterize their genomic features and construct a comprehensive phylogenetic tree. Our aim was to elucidate the phylogenetic relationships within the genus and evaluate interspecific affinities among the subgenera. We also addressed the taxonomic ambiguity surrounding *A. bouchonii* and *A. powellii* to determine their distinct species within the genus. Chloroplast genome sizes ranged from 149,949 to 150,818 bp, with GC content varying between 36.52% and 36.63%. Comparative structural analyses confirmed highly conserved quadripartite structures, gene content, and organization, comprising 87 protein-coding genes, 37 tRNAs, and 8 rRNAs. Repeat and codon usage analyses revealed conserved repeat patterns and a preference for codons ending in A or U. Selection pressure analysis indicated a predominantly purifying selection, with *matK* showing signs of positive selection, particularly in *A. spinosus*. Phylogenetic analysis of 80 protein-coding genes confirmed the monophyly of subgenus *Amaranthus* but found *Alberisa* and *Acnida* to be paraphyletic. Despite their morphological similarity, *A. bouchonii* and *A. powellii* were placed in separate clades within subgenus *Amaranthus*, with *A. bouchonii* clustering with *A. retroflexus*, and *A. powellii* aligning with the *A. hybridus* complex. Additionally, we identified 16 variable regions as potential molecular markers for species identification. Our study provides the most comprehensive *Amaranthus* chloroplast genome dataset to date, offering new insights into its evolutionary relationships and valuable genomic resources for taxonomy, germplasm management, and invasive risk assessment.

## 1. Introduction

The genus *Amaranthus* consists of approximately 70–80 species, widely distributed across tropical, subtropical, and temperate regions of the world [1]. Over ten taxa of this genus exhibit diverse functional roles, including use as pseudocereals (e.g., *A. cruentus, A. hypochondriacus*, and *A. caudatus*) and traditional leafy vegetables (e.g., *A*. *tricolor*, *A. cruentus*) [1,2,3]. Rich in bioactive compounds with medicinal properties, the *Amaranthus* species are highly nutritious and demonstrate strong ecological adaptability, including drought resistance and heavy metal tolerance, making them promising candidates for climate-resilient food crops [4,5,6,7,8,9,10]. However, the genus also includes several highly invasive species that pose significant ecological and economic threats. Traits such as prolific seed production, efficient dispersal mechanisms, and a strong reproductive capacity enable these species to rapidly proliferate and dominate, outcompeting agricultural crops. Globally, around 20 *Amaranthus* species are recognized as pervasive weeds [11], including *A. palmeri*, *A. powellii*, *A. retroflexus*, *A. spinosus*, *A. tuberculatus*, and *A. viridis*, according to the USDA Plants Database [12]. These species cause considerable yield losses, particularly in cereals and vegetables, which are vital to both agricultural and non-agricultural sectors [13,14,15].

The genus *Amaranthus* has been divided into three subgenera—*Amaranthus*, *Acnida,* and *Albersia*—based on morphological features such as inflorescence position, perianth segment number, fruit dehiscence mode, and sexual characteristics [2,16]. However, the relationship among these subgenera remains unclear, and species delimitation is often ambiguous due to extensive interspecific hybridization, high phenotypic plasticity, and inconspicuous morphological traits [11]. This has led to ongoing debates regarding the classification of certain species, particularly *A. bouchonii* and *A. powellii*, which have been disputed for decades due to conflicting biological and morphological evidence [1,17,18]. For instance, isoenzymes and morphological studies suggest that *A. bouchonii* should be considered a distinct taxon closely related to *A. powellii* [17], while others proposed it as a subspecies of *A. powellii* (i.e., *A.*
*powellii* subsp. *bouchonii*) based on differences in fruit, seed, and inflorescence structure [2]. More recent research highlights distinct chromosomal and genomic differences between the two species, further complicating the issue [18]. Compounding this uncertainty, *A. powellii* poses serious ecological and agricultural threats, including crop yield loss, allelopathic chemicals, and livestock toxicity [19,20]. Given both the taxonomic ambiguity and its ecological impact, further phylogenetic research into *Amaranthus* is urgently needed to clarify the relationship between *A. powellii* and *A. bouchonii*.

Molecular phylogenetic analysis can resolve evolutionary relationships that are difficult to detect through morphology alone, providing a more accurate framework for plant classification [21]. Such studies have been essential in other plants [22,23,24], where phylogenetic reclassification based on molecular data has led to a better understanding of evolutionary relationships and a more accurate taxonomic framework [25]. In *Amaranthus*, improving the phylogenetic resolution through molecular data will not only clarify the taxonomic relationships between *A. powellii* and *A. bouchonii* but also are essential to clarify relationships within the three subgenera of *Amaranthus* and assess the interspecific relationships.

Previous molecular phylogenetic studies have employed a variety of markers, including chloroplast DNA (e.g., *matK*, *trnL*), nuclear internal transcribed spacers (ITSs), low-copy nuclear genes, and bi-allelic SNPs [26,27,28,29,30,31]. While some studies support subgeneric classifications based on morphology [16,30], others highlight significant limitations of morphology-based taxonomy, such as phenotypic plasticity, morphological convergence, and frequent hybridization, which can obscure evolutionary relationships [6,27,29,31,32]. Additionally, some molecular studies have faced challenges, including insufficient sample sizes, unclear species identification, and the use of low-resolution genetic markers, leading to conflicting phylogenetic inferences [32].

Chloroplast genome studies have helped in clarifying the evolutionary relationships within *Amaranthus* [31]. Key features of *Amaranthus* chloroplast genomes, including genome size (150–151 kb), GC content, and the number of protein-coding genes (CDSs), are highly conserved across the genus, with studies confirming their consistency among species [31,32,33]. Moreover, regions such as the *trnLUAG*-*ccsA*-*ndhD* intergenic spacer exhibit high divergence across *Amaranthus* species, making them promising candidates for molecular markers or DNA barcoding [32]. These characteristics highlight the potential of chloroplast genomics for phylogenetic and taxonomic research in *Amaranthus*. Despite significant progress, challenges persist in resolving the phylogenetic relationships within *Amaranthus*, including limited taxon sampling and insufficient resolution across subgenera.

In this study, we sequenced and assembled the chloroplast genomes of *A. powellii* and *A. bouchonii* for the first time, contributing valuable genomic data to the *Amaranthus* phylogenetic framework. In addition, we reviewed and reannotated eight publicly available chloroplast genome datasets from the NCBI Sequence Read Archive (SRA) and incorporated 17 publicly available genomes from the GenBank database, resulting in a total of 27 complete *Amaranthus* chloroplast genomes. We then conducted a comprehensive analysis of the chloroplast genome characteristics across these species. The main objectives of this study are the following: (1) To construct a phylogenetic framework for the *Amaranthus* species based on chloroplast genome data, revealing their evolutionary relationships and examining the characteristics of their chloroplast genomes. (2) To identify potential divergent hotspots within the chloroplast genomes that could serve as molecular markers for species delimitation. (3) To resolve the taxonomic relationship between *A. powellii* and *A. bouchonii* using molecular evidence. By focusing on chloroplast genomes, this work provides deeper insights into the taxonomy and evolution of *Amaranthus* and offers valuable molecular tools for future research and practical applications.

## 2. Results

### 2.1. Genome Size and Basic Structural Features

The chloroplast genomes of *A. bouchonii* and *A. powellii* exhibit a certain degree of difference in sequence length and GC content (Appendix A). The chloroplast genome of *A. powellii* is the longest among all *Amaranthus* species analyzed, with genome lengths of 150,818 bp and 150,775 bp in two sampled individuals, *A. bouchonii* has a relatively shorter genome at 150,721 bp. The GC content differed slightly between the two species, with *A. bouchonii* at 36.59% and *A. powellii* at 36.56%. Across all 27 *Amaranthus* species with published chloroplast genomes, a typical quadripartite circular structure is consistently observed (Figure 1). Genome sizes range from 149,949 bp in *A. polygonoides* to 150,818 bp in *A. powellii*, reflecting a maximum length difference of 869 bp. The GC content varies slightly, ranging from 36.52% to 36.63% (Appendix A). Despite these variations, all species share a conserved set of 132 genes, including 87 CDSs, 37 tRNAs, and 8 rRNAs, showing highly conserved positions and copy numbers across the genus (Appendix A). These genes are classified into four functional categories: 73 self-replication genes, 44 photosynthetic genes, 6 other functional genes, and 9 genes of unknown function. Notably, 18 genes, including *rrn4.5*, *trnL-CAA*, and *ycf1*, occur in duplicate copies, while the remaining genes exist as single-copies (Appendix A).

### 2.2. Analysis of Scattered Repeat Sequences and Simple Repeat Sequences

The microsatellite count in *A. powellii* is 81, which exceeds that of *A. bouchonii* by 4 (77), a difference encompassing 3 single-nucleotide microsatellites and 2 dinucleotide microsatellites. Both species exhibit 21 palindromic sequences; however, *A. powellii* possesses 19 forward repeats compared to *A. bouchonii*’s 17. In contrast, *A. bouchonii* has four reverse repeats, whereas *A. powellii* only contains one. Neither species was found to have any complementary repeat sequences (Appendix A). In the 27 species of *Amaranthus*, a total of 2082 SSR loci were identified. The number of SSRs across different *Amaranthus* species does not exhibit significant variation, ranging from 71 to 85. These include 1142 mononucleotides, 200 dinucleotides, 103 trinucleotides, 302 tetranucleotides, and 73 pentanucleotides (prevalent only in *A. cruentus*, *A. hybridus*, *A. hypochondriacus*, *A. cannabinus*, and *A. powellii*). There are also 30 hexanucleotides, with only 2 found in *A. spinosus*, *A. dubius*, and *A. powellii*, while the remaining species contain only 1. The count of interspersed repeat sequences varies between 21 and 24, totaling 1043. This includes 600 palindromic repeats, 416 forward repeats, 26 reverse repeats, and 1 complementary repeat (unique to *A. blitoides*) (Figure 2, Appendix A).

### 2.3. Codon Usage Bias Patterns

Both *A. bouchonii* and *A. powellii* demonstrated a codon preference, with the most frequently utilized codon in *A. bouchonii* being AUU (ILE), appearing 1161 times, while UGC (CYS) was the least common, occurring only 74 times. Similarly, in *A. powellii*, the most frequently used codon was also AUU (ILE), appearing 1160 times, with UGC (CYS) again being the least common, appearing only 74 times. These findings align with the codon selection preferences observed in other species within the genus *Amaranthus* (Appendix A). The codon count in 27 species of *Amaranthus* varies from 26,215 (*A. polygonoides*) to 27,069 (*A. hybridus*), with no significant differences observed among the species. Both Tryptophan (Try) and Methionine (Met) are encoded by a single codon, while other amino acids are encoded by between two and six synonymous codons. The relative synonymous codon usage (RSCU) can reveal any bias towards a specific codon usage and its intensity: an RSCU greater than 1 indicates a higher frequency of codon usage; an RSCU equal to 1 suggests no preference; and an RSCU less than 1 denotes a lower frequency. There are 29 codons with an RSCU value exceeding 1, suggesting a higher usage frequency. Notably, the third base of these 29 codons is either A or U, with the sole exception being the UU encoding Leucine (Leu), which has a C at its third position. This pattern suggests that the 27 species of *Amaranthus* exhibit a preference for codons ending in A or U. Conversely, there are 30 codons with an RSCU value below 1, indicating a lower usage frequency. The RSCU values for both Methionine (Met) and Tryptophan (Trp) are exactly 1, indicating no specific bias towards their codon usage (Figure 3, Appendix A).

### 2.4. Selection Pressure Patterns in Amaranthus Chloroplast Genes

The selection pressure analysis, using *A. powellii* as the reference sequence, revealed that the majority of genes (71) in *A. bouchonii* exhibited neutral evolution. In contrast, only nine genes were subjected to negative selection: *accD*, *atpA*, *atpE*, *ndhD*, *psaA*, *rpl22*, *rpoC2*, *rps15*, and *ycf1* (Appendix A).

When analyzing 27 species within the *Amaranthus* genus, most genes were found to be under negative selection. However, eight genes—*ccsA*, *atpA*, *clpP*, *matK*, *rpcL*, *rpl16*, *rps18*, and *ycf2*—exhibited positive selection in specific species. Notably, in the *matK* gene, positive selection was observed in ten species: *A. australis*, *A. blitoides*, *A. blitum*, *A. cannabinus*, *A. capensis*, *A. crispus*, *A. deflexus*, *A. dubius*, *A. spinosus*, and *A. standleyanus*, with *A. spinosus* exhibiting the highest selection intensity at 2.04584. Additionally, genes such as *aacD*, *accA*, *clpP*, *rplC*, *rpl16*, *rps18*, and *ycf2* demonstrated positive selection in individual or a limited number of species, although with lower selection intensity. Conversely, genes such as *petG*, *petL*, *psaL*, *psbD*, *psbE*, *psbF*, *psbH*, *psbJ*, *psbK*, *psbM*, *psbN*, and *psbT* were found to be under neutral selection across the *Amaranthus* genus (Figure 4, Appendix A).

### 2.5. Structural Conservation and Variation

The mVISTA results indicated a high degree of conservation, with no rearrangements or inversions observed in these genomes (Appendix A). Variation frequency was found to be higher in non-coding regions compared to coding regions. Both the LSC and SSC regions exhibited greater variations, while the IR region demonstrated relatively smaller variations, suggesting its relative conservation. Collinearity analysis revealed no gene rearrangement or inversion, further supporting the relative conservation of the chloroplast genomes among the 27 *Amaranthus* species (Appendix A). An examination of the LSC/IRb (JLB), IRb/SSC (JSB), SSC/IRa (JSA), and IRa/LSC (JLA) boundaries of the 27 chloroplast genomes revealed that all 27 genomes had identical types of genes distributed at these boundaries, with differences only in the position and size of the genes (Appendix A). At the JLB boundary, the genes *rpl22* and *rps19* are distributed, with all species’ *rps19* genes crossing the JLB boundary and all *rpl22* genes located in the LSC. At the JSB boundary, the genes *ycf1* and *ndhF* are distributed, with both *ycf1* and *ndhF* genes crossing the JSB boundary. At the JSA boundary, the genes *ycf1* and *trnN* are distributed, with the *ycf1* gene spanning across the JSA boundary and all *trnN* genes located in the IRa region. The JLA boundary is situated between the *rpl2* gene and the *trnH* gene.

### 2.6. Nucleotide Diversity and Highly Variable Regions

The Pi values for CDSs range from 0 to 0.0128, while the intergenic regions show a broader range, from 0 to 0.03256 (Figure 5, Appendix A). Notably, the Pi value for the intergenic regions is 2.5 times higher than that of the coding regions (0.03256/0.0128 = 2.5), indicating that nucleotide polymorphism in the intergenic regions significantly exceeds that in the coding regions and exhibits heterogeneous variation. Genes with a Pi value greater than 0.010 are considered highly variable. The genes and intergenic regions with the highest Pi values, listed in descending order, include the following: *atpA*+—*atpF* exon2−, *rpl22*+—*rps19*−, *psbN*−—*psbT*−, *ccsA*+—*rpl32*−, *rpl22*, *ndhD*+—*psaC*−, *ndhF*+—*rpl32*+, *ccsA*−—*ndhD*−, *rpl33*−—*rps18*+, *psaJ*−—*rpl33*+, psbI+—*psbK*−, *psaI*−—*ycf4*+, *ndhE*−—*psaC*+, *rps15*+—*ycf1*−, *rps4*−—*ycf3* exon1+, *ndhA* exon2−—*ndhI*+. Furthermore, these highly variable genes and intergenic regions predominantly localize in LSC and SSC, with a diminished variation observed in the IR region.

### 2.7. Phylogenetic Relationships of Amaranthus

The phylogenetic results indicate that, within the Amaranthaceae, *Celosia* is sister to *Amaranthus* (Figure 6). Within *Amaranthus*, the subgenus *Albersia* (highlighted in blue) does not form a single monophyletic group, but instead splits into three distinct clades: A, B, and C. Clade A, comprising seven species (*A. blitum*, *A. capensis*, *A. tricolor*, *A. crispus*, *A. standleyanus*, *A. deflexus*, and *A. viridis*), received strong support (bootstrap support [BS] = 100) and, together with *A. albus*, *A. blitoides*, *A. polygonoides*, Clades D (two species of subgen. *Acnidia*), and Clade G, forms a poorly resolved polytomy. Within Clade G, Clade E (the rest of subgen. *Acnidia*) and Clade F (subgen. *Amaranthus*) are sister to each other with strong statistical support (BS = 100). Within Clade F, *A. bouchonii* and *A. retroflexus* form a clade that is sister to the remaining members of the subgenus (BS = 100). *A. dubius* and *A. spinosus* form a secondary diverging group (BS = 100), which in turn is sister to a clade comprising the members of the *A. hybridus* complex (*A. powellii*, *A. hybridus*, *A. cruentus*, and *A. hypochondriacus*, BS = 100).

## 3. Discussion

### 3.1. Molecular and Morphological Perspectives on the Taxonomy of A. powellii and A. bouchonii

Since species within *Amaranthus* exhibit high morphological similarity, taxonomic uncertainties often arise, particularly in distinguishing *A. powellii* and *A. bouchonii*. *A. powellii, native* to southwestern North America, has expanded its range across multiple continents, including North America, South America, Europe, Asia, and Australia. Morphologically, it is characterized by erect, unbranched, or sparsely branched inflorescences, fruits that dehisce in a ring-like manner (approximately twice as long as wide), and seeds with faint grooves extending one-third to one-half of their length, with nearly smooth edges [2,18,19,20]. In contrast, *A. bouchonii,* first described by Thellung [34] based on European specimens, is distinguished by its non-dehiscent fruit, a trait central to its taxonomic definition [34]. However, morphology alone is insufficient for resolving taxonomic ambiguities in the genus.

This study provides new insights into their taxonomic status based on phylogenetic analysis and comparative chloroplast genome data. Our phylogenetic results show that *A. powellii* clusters with the *A. hybridus* complex, while *A. bouchonii* forms a sister clade to *A. retroflexus* (Figure 6, BS = 100), a result consistent with previous findings based on low-copy nuclear genes and plastid fragments [30], and is consistent with Thellung’s classification, which originally considered *A. bouchonii* a separate species from *A. powellii* [34]. The close phylogenetic relationship between *A. bouchonii* and *A. retroflexus* suggests a shared evolutionary history, yet their morphological differences—particularly the stable ring-like dehiscence of *A. retroflexus* fruits—support their recognition as separate species [20].

Beyond phylogenetics, our comparative genomic analysis further supports their taxonomic separation. Although *A. powellii* and *A. bouchonii* share a largely conserved chloroplast genome structure, differences in SSRs, reverse repeat sequence distributions, and variable genomic regions provide additional molecular evidence for their distinction (Appendix A). Furthermore, their significant ecological differences reinforce this separation: in Europe, *A. bouchonii* primarily colonizes riverbanks as pioneer species, whereas *A. powellii* thrives in abandoned fields and wastelands as an invasive weed [2,20]. These combined molecular and ecological findings substantiate their classification as separate species.

Despite these findings, identifying *A. bouchonii* remains complex, particularly because of variations in its fruit dehiscence mechanism. Since fruit dehiscence is a key taxonomic trait within Amaranthaceae, Moquin et al. have proposed subgroup classifications (e.g., *Ambogayna*, *Euxolus*) based on this feature [34]. In Europe, the non-dehiscent fruit trait of *A. bouchonii* is relatively stable. However, in North America, populations of *A. bouchonii* show substantial variability [2,18,31], with some specimens resembling *A. powellii* and others displaying mixed traits, such as ring-like or irregular dehiscence, akin to *A. hybridus* [2]. From the perspective of geographic distribution, the vast majority of reports regarding *A. bouchonii* originate from Europe, whereas populations in the Americas are far less common. While most species within *Amaranthus* are believed to have originated in the Americas, there is debate that *A. bouchonii* may have originated in Europe [18], or alternatively, that it diverged from an ancestral species introduced from the Americas to Europe, where it subsequently differentiated [34]. These viewpoints highlight the complexity of its evolutionary history and emphasize the need for additional investigation. To accurately define *A. bouchonii*, further research is required, including broader sampling from both its presumed native regions and other distribution areas. Detailed analyses of its original type specimens are also essential. Such studies will help elucidate its evolutionary relationships and refine its taxonomic status.

### 3.2. Genomic Organization and Conserved Features of Amaranthus Chloroplasts

The chloroplast genome serves as a vital resource for investigating plant evolution, systematic development, and genetic diversity [35,36,37,38]. The *Amaranthus* chloroplast genomes exhibit a highly conserved structure, and might reflect evolutionary constraints that maintain essential physiological functions. It follows a typical quadripartite organization, comprising an LSC region, an SSC region, and two IR regions. Across all analyzed species, the genome size (149,949–150,818 bp) and GC content (36.52–36.63%) remain stable, consistent with previous reports [31,33,39]. The conserved number of 87 protein-coding genes, 37 tRNAs, and 8 rRNAs highlights strong structural uniformity within the genus, suggesting that chloroplast genomes in *Amaranthus* may be under selective pressure to maintain genomic stability, particularly for photosynthetic efficiency and self-replication processes [26,40,41].

Further supporting this structural stability, sequence alignment and synteny analysis detected no genome rearrangements or inversions, while the IR region boundaries exhibited minimal variation across species. The limited expansion and contraction of IRs suggest that *Amaranthus* plastomes possess robust repair and stabilization mechanisms, which help maintain genome integrity and prevent large-scale structural modifications. These patterns are consistent with other genera in Amaranthaceae, such as *Chenopodium* and *Alternanthera* [42,43]. However, while coding regions remain highly conserved, the intergenic spacers in the LSC and SSC regions exhibit greater sequence divergence, serving as potential hotspots for evolutionary change within *Amaranthus*, consistent with patterns reported in other angiosperm lineages [36,39,44,45,46,47,48,49].

Repetitive elements, such as SSR markers, characterized by their high polymorphism, codominant inheritance, and ease of detection, are widely used in genetic research [50,51,52]. Additionally, dispersed repeats and SSRs play key roles in genome evolution, rearrangement, and stability [53,54,55,56]. In *Amaranthus*, we identified a total of 2082 SSR loci and 1084 dispersed repeat sequences across 27 species, with a highly consistent distribution pattern across the chloroplast genome. This pattern aligns with previous findings on repetitive elements in *Amaranthus* chloroplast genomes, further reinforcing the evolutionary stability of these sequences [31,32]. Despite their overall conservation, these SSR markers exhibit significant potential for polymorphism in *Amaranthus* species, and the high variability in SSR types and numbers among different species (Appendix A) makes them valuable for germplasm identification and molecular breeding applications [55,56,57,58]. The total number of SSRs ranges from 71 (*A. cruentus*) to 85 (*A. blitoides*), with mono-nucleotide repeats being the most abundant, comprising 69.30% (1499 out of 2163 total SSRs), whereas hexa-nucleotide repeats are the least common, accounting for only 1.48% (32 out of 2163). Compared to previous studies, the SSR abundance in *Amaranthus* chloroplast genomes is moderate, as other plant groups have shown either higher or lower SSR densities, depending on their evolutionary lineage and genome composition [26,32,50,52]. These findings suggest that while the observed SSR variation in *Amaranthus* plastomes is relatively limited, it remains sufficiently informative for genetic diversity assessments, population structure analyses, and phylogenetic studies.

The codon preference analysis revealed that *Amaranthus* chloroplast genomes share similarities with those of other angiosperms, exhibiting a strong preference for codons ending in A or U, with leucine being the most frequently used amino acid (10.73%) [48,59,60,61]. This conserved bias reflects the mutational and selective pressures acting on chloroplast genomes across lineages. While most genes in *Amaranthus* chloroplast genomes are under purifying selection, *matK* showed signs of positive selection in species such as *A. spinosus*. Its elevated selection intensity (2.04584) suggests a potential functional role in adaptation to arid or saline–alkaline environments [62,63], although further experimental validation is required to confirm its functional role in environmental stress adaptation [41,64].

Despite low overall nucleotide diversity (Pi values), 16 highly variable regions were identified, primarily in the LSC and SSC regions. These loci provide valuable molecular markers for phylogenetic and population genetic studies. The nucleotide diversity of *Amaranthus* chloroplast genomes is relatively lower than that observed in various genera both in Dicots and Monocots, for example, *Pulsatilla*, *Byrsonima,* and *Aldama* in Dicots, and *Agropyron* and *Lilium* in Monocot [46,47,65,66], indicating a conserved evolutionary pattern within the genus.

The high degree of conservation in *Amaranthus* chloroplast genomes underscores their evolutionary stability within Amaranthaceae and their utility for comparative studies across plant families [32,46]. However, the stability also highlights the need to incorporate nuclear and mitochondrial genomes to complement chloroplast data in resolving complex evolutionary relationships.

### 3.3. Evolutionary Relationships of the Chloroplast Genome in the Genus Amaranthus

*Amaranthus* presents taxonomic challenges due to several complicating factors, including limited distinguishing morphological characters, the small and challenging nature of diagnostic parts, a broad geographical distribution, and extensive hybridization among species. These factors have led systematists to describe *Amaranthus* as a ‘difficult’ genus [2,11]. Historically, certain species were recognized as separate genera, particularly the dioecious and monoecious species with dehiscent or indehiscent fruits. However, Sauer [67] and Robertso [68] later consolidated these groups into *Amaranthus*, currently recognized as three subgenera: *Acnida*, *Amaranthus*, and *Albersia* [2,16]. Subgenus *Acnida* comprises dioecious species, while *Amaranthus* and *Albersia* split monoecious species based on inflorescence position, tepal number, and fruit dehiscence [16]. Nonetheless, multiple studies have suggested that taxonomic classifications based solely on morphological traits may not accurately reflect the evolutionary history of the genus [69,70]. In our study, our phylogenetic analysis involved a dataset of 27 species from three subgenera of *Amaranthus*, providing a comprehensive phylogenetic framework. The results suggest that, based on our current sampling, neither *Albersia* (highlighted in blue) nor *Acnida* (highlighted in red) forms monophyly (Figure 6). This finding aligns with earlier chloroplast genome-based studies, which classified *A. albus*, *A. blitoides*, and *A. polygonoides* (from subgens *Albersia*) into the paraphyletic Galápagos clade [30,31,32]. Phylogenetic studies using limited gene datasets, such as ITS, ALS, and *rpoC2*, initially supported their inclusion within subgenus *Albersia* with moderate to high support [30,71]. However, analyses incorporating more species and gene sequences consistently revealed divergent clades, further challenging the monophyly of *Albersia* [30,32,71]. Phylogenetic analyses based on ALS and ITS positioned subgenus *Albersia* as a basal clade, although the boundary between *Amaranthus* and *Acnida* remained unclear [71]. Conflicting topologies in phylogenetic trees based on different datasets highlight challenges arising from low-resolution molecular markers, interspecific hybridization, gene duplication, incomplete lineage sorting, and gene flow, suggesting a more complex evolutionary history for the genus [15,30,72,73,74,75,76]. Resolving these ambiguities will require broader genomic data integration, including nuclear and mitochondrial genome analyses [77,78,79].

*Acnida* consists of nine dioecious species native to North America [16,32,69]. Our phylogenetic analysis corroborates the findings of Raiyemo et al. [32], who provided strong support for the main branch of the dioecious species clade (Clade E) and confirmed the sister relationship between subgenera *Acnida* and *Amaranthus* (BS = 100). Raiyemo et al. [32] also hypothesized that chloroplast genomes, due to their non-recombining and uniparental inheritance, might reflect hybridization or chloroplast capture events, potentially contributing to the divergence of the *A. australis* + *A. cannabinus* lineage within the *Acnida* + *Amaranthus* clades. For example, *A. arenicola,* a dioecious species, exhibits morphological traits overlapping with *A. palmeri* (female flowers with five perianth segments) and *A. tuberculatus* (short bracts and long spikes). Stetter and Schmid [29] placed *A. arenicola* and *A. palmeri* together based on genotyping by sequencing data. However, Xu et al. [71], using ALS domain analyses, suggested closer relationships between *A. arenicola* and *A. palmeri*, while ITS and other ALS domains placed it closer to *A. tuberculatus*. Waselkov et al. [30] concluded that *A. arenicola* is most closely related to *A. tuberculatus* based on nuclear and chloroplast data, a result consistent with our phylogenetic findings.

In subgenus *Amaranthus*, *A. bouchonii* and *A. retroflexus* form a highly supported sister clade (BS = 100), contrasting with earlier research indicating that *A. bouchonii* and *A. powellii* first form a monophyletic group, which then clusters with the sister clade of *A. retroflexus* and *A. wrightii* [30]. In this study, two *A. powellii* individuals cluster with the other members of the *A. hybridus* complex (*A. hybridus*, *A. cruentus*, and *A. hypochondriacus*), forming a distinct monophyletic group (BS = 100) based on chloroplast genome data. These findings further confirmed the potential discrepancies between nuclear and chloroplast phylogenies, highlighting the complex evolutionary dynamics of *Amaranthus* [26,71].

## 4. Materials and Methods

### 4.1. Samples Collection, DNA Extraction, and Sequencing

In this study, we employed a high-throughput sequencing approach to generate chloroplast genome sequences from specimens collected at different locations. These include one *A. bouchonii* sample (Yueyang, China, specimen number CSH0141775) and two *A. powellii* samples (Qinhuangdao, China, specimen numbers CSHO138866 and CSHO138875); all samples are recent introductions to China and are preserved in the Shanghai Chenshan Botanical Garden Herbarium (CSH). Additionally, we sourced SRA raw sequencing data for 8 *Amaranthus* species and 17 chloroplast genomes from the NCBI database. A total of 27 *Amaranthus* species were analyzed in the comparative chloroplast genomic study. To further explore the evolutionary relationships of *Amaranthus* within the Amaranthaceae, we included chloroplast genomes from 11 additional genera in the family, as well as an outgroup species, *Phaulothamnus spinescens* (Achatocarpaceae), in the phylogenetic analysis. Detailed information on all data sources can be found in Appendix A.

Healthy, fresh leaves were collected from *A. bouchonii* and *A. powellii* and preserved in silica gel at −20 °C. Genomic DNA was extracted using a modified CTAB method [80]. The quality and concentration of the extracted DNA were assessed using a NanoDrop 2000 spectrophotometer (Thermo Fisher Scientific, Waltham, MA, USA), while agarose gel electrophoresis was employed to evaluate the DNA integrity and contamination. High-quality DNA was fragmented into random pieces via ultrasonication, followed by end-repair and the addition of an ‘A’ nucleotide to the 3′ ends. Sequencing adapters were then ligated, and magnetic beads were used to select fragments approximately 400 bp in length. The enriched fragments were amplified by PCR to construct a sequencing library, which was subsequently assessed for quality. Libraries that met quality standards were sequenced on the Illumina HiSeqTM platform (Majorbio, Shanghai, China), using a paired-edn 150 bp sequencing strategy, yielding a total read length of 300 bp.

### 4.2. Chloroplast Genomes De Novo Assembly and Annotation

Raw image data files from high-throughput sequencing were subjected to base calling and converted into raw reads [81]. Low-quality data were filtered using the quality control software fastp v0.19.6 [82] to obtain clean reads. In addition, raw sequencing data downloaded from the SRA were processed using the same pipeline to ensure consistency. The GetOrganelle v1.7.7.0 [83] was employed for de novo assembly of the chloroplast genomes, while PGA [84] was used for the annotation of both newly assembled and NCBI-sourced *Amaranthus* chloroplast genome sequences. Manual checking and further adjustments were performed using Geneious v9.0.2 [83] to ensure accuracy and completeness of annotations.

### 4.3. Analysis of Simple Repeats and Interspersed Repeats

The online tool, MISA-web [85], was employed to identify and enumerate Simple Sequence Repeats (SSRs) in the chloroplast genome sequences of 27 *Amaranthus* species. The parameters were configured as follows: mononucleotides with a minimum of ten repeat units, dinucleotides with a minimum of five repeat units, trinucleotides with a minimum of four repeat units, tetranucleotides, pentanucleotides, and hexanucleotides with a minimum of three repeat units. Additionally, the online tool REPuter (https://bibiserv.cebitec.unibielefeld.de/reputer, accessed on 14 October 2024) was used to count interspersed repeats. These include forward repeats, reverse repeats, palindromic repeats, and complementary repeats, with the minimum length (minimal repeat size) set at 30 bp and the maximum Hamming distance set at 3.

### 4.4. Codon Usage Bias Analysis

CDSs were extracted from the genome utilizing Phylosuite [86]. The relative synonymous codon usage (RSCU) and effective number of codons (ENCs) analysis for *Amaranthus* chloroplast CDSs was conducted using CodonW v1.4.4 [86], with the aim of investigating the patterns of codon usage and synonymous codon usage (RSCU) values. An RSCU value > 1 indicates that the codon was preferentially used by amino acids, whereas an RSCU value < 1 indicates the opposite trend.

### 4.5. Selection Pressure Analysis

In genetics, the ratio of nonsynonymous (Ka) to synonymous (Ks) substitution rates (ω) is frequently used to evaluate the presence of selection pressure on CDSs [87]. In this study, we used a Perl script to extract CDSs from the chloroplast genome of *Amaranthus*, used Geneious to extract CDSs, and calculated the Ka, Ks, and Ka/Ks values for each plastid gene with a KaKs_calculator [88]. The evolutionary process of genes, i.e., positive selection, purified selection, or neutral selection, is Ka > Ks (ω > 1), Ka < Ks (ω < 1), or Ka = Ks (ω = 1), respectively.

### 4.6. Comparative Analysis of Genome Structure

To detect the variable regions in *Amranthus*, the mVISTA program [89] was used in the Shuffle-LAGAN mode with *A. blitum* as a reference for chloroplast genome sequence comparison. The progressive mauve algorithm [90] was used to compare the locally collinear blocks (LCBs) and genomic rearrangements among the chloroplast genomes. The comparative lengths and boundaries of the quadripartite structures in the chloroplast genome were drawn by CPJSdraw [91], the progressive mauve algorithm, and a manually drawn chloroplast genome gene order plot.

### 4.7. Nucleotide Polymorphism Analysis

Nucleotide polymorphism (Pi) in *Amaranthus* was calculated using DnaSP v6 [92] for coding and non-coding regions, treating each species as a distinct unit of comparison. We followed the standard equation for nucleotide diversity (Pi), given by the following:$$\pi =\frac{1}{L}\times \frac{\sum_{i<j}I_{ij}}{\frac{n\left(n-1\right)}{2}}$$
where *L* is the length of the sequence. *I_ij_* represents the differences between the *i*-th and *j*-th sample, typically calculated based on the different bases in each pairwise comparison. n is the number of samples. The calculations were performed using a sliding window approach (window size = 1000 bp, step size = 500 bp), and the results were visualized in R v4.2.1 using ggplot2 [93].

### 4.8. Phylogenetic Analysis

The analysis included 38 species from 12 genera in the Amaranthaceae family, plus an outgroup species. CDS sequences were carefully extracted from GenBank files and linked using two python scripts (https://github.com/Kinggerm/PersonalUtilities, accessed on 10 October 2024). These sequences were then aligned using MAFFT v7.50 [94], making the matrix length 64,128 bp for analysis. The ModelFinder module of IQ-Tree v1.6.8 [95] was used to determine the best evolutionary model for target data. ML analysis was performed based on the above parameters and models, and repeated 5000 times with the number of threads specified as ‘AUTO’ for the construction of maximum likelihood (ML) phylogenetic trees. The generated phylogenetic trees were annotated using the R package ggtree [96].

## 5. Conclusions

This study presents the first complete chloroplast genome sequences of *A. powellii* and *A. bouchonii*, confirming their status as distinct species based on phylogenetic analysis and comparative analysis of chloroplast genome characteristics. By incorporating previously published chloroplast genome data, we expand the analysis to include 27 *Amaranthus* species, providing a comprehensive overview of chloroplast genome characteristics within the genus, further confirming their high level of conservation. Despite the challenges posed by morphological similarities and frequent interspecific hybridization, the phylogenetic tree constructed in this study offers valuable insights into the evolutionary relationships among *Amaranthus* species. However, further research is required, including integrative phylogenetic approaches that combine morphological and molecular data, along with biogeographical and macroevolutionary analyses. Such studies are essential to refine our understanding of the evolutionary relationships and classification of *Amaranthus* species, elucidate the timing of key speciation events, and explore the underlying mechanisms driving these processes. In addition, addressing the observed topological inconsistencies across different datasets is critical. These studies will help resolve gene tree conflicts and other evolutionary complexities, contributing to a more comprehensive understanding of the genus’ evolutionary history.

## Figures and Tables

**Figure 1 plants-14-00649-f001:**
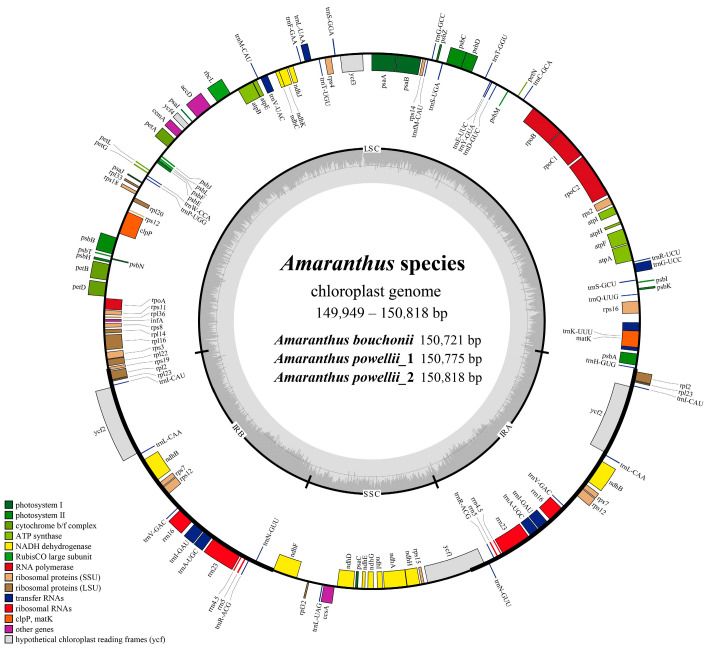
Genetic physical maps of *Amaranthus* species. Genes located on the outer circle are transcribed clockwise, while those in the inner circle are transcribed in a counterclockwise direction. The dark gray area represents GC content, while the light gray area represents AT content. Gene functions are color-coded.

**Figure 2 plants-14-00649-f002:**
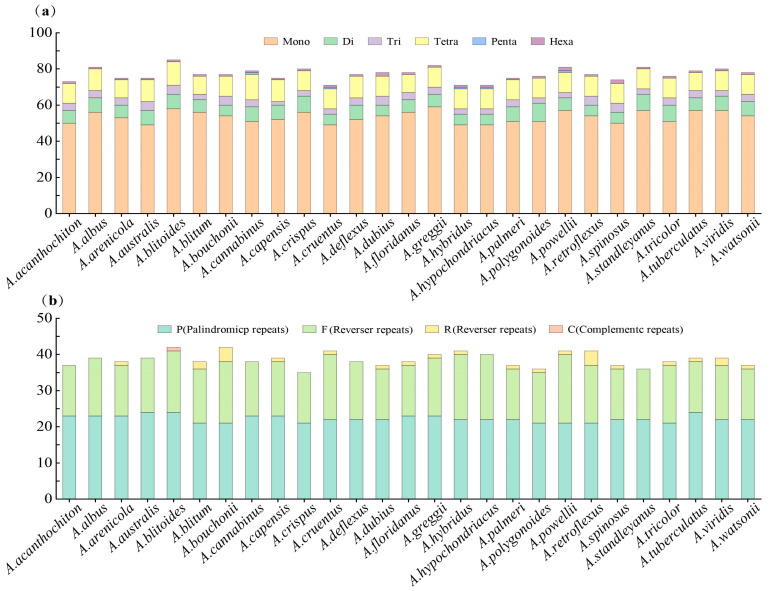
(**a**) The number of microsatellites across 27 species of the *Amaranthus* genus, with different colors representing different types of microsatellites. (**b**) The number of inverted repeats across 27 species of the *Amaranthus* genus, with different colors representing different types of inverted repeats.

**Figure 3 plants-14-00649-f003:**
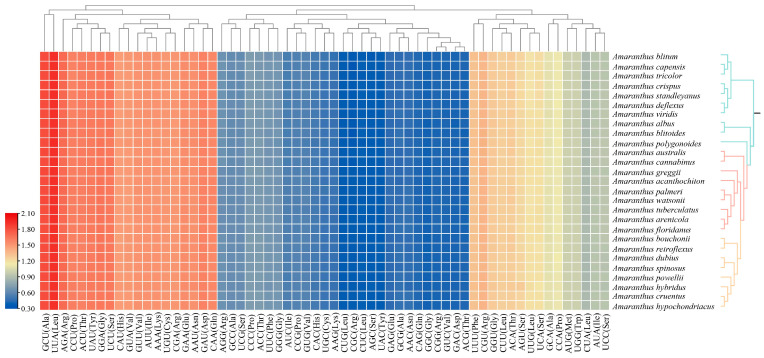
Codon usage preference analysis of *Amaranthus* species. The color gradient ranges from blue to red, indicating increasing codon preference. The phylogenetic tree on the right was inferred from the 80 CDS dataset in subsequent analysis.

**Figure 4 plants-14-00649-f004:**
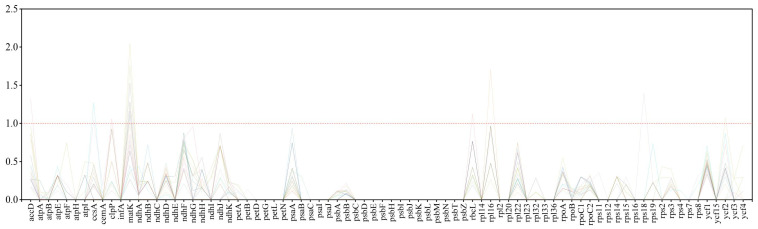
Ka/Ks analysis of the 80 CDS regions in *Amaranthus* species.

**Figure 5 plants-14-00649-f005:**
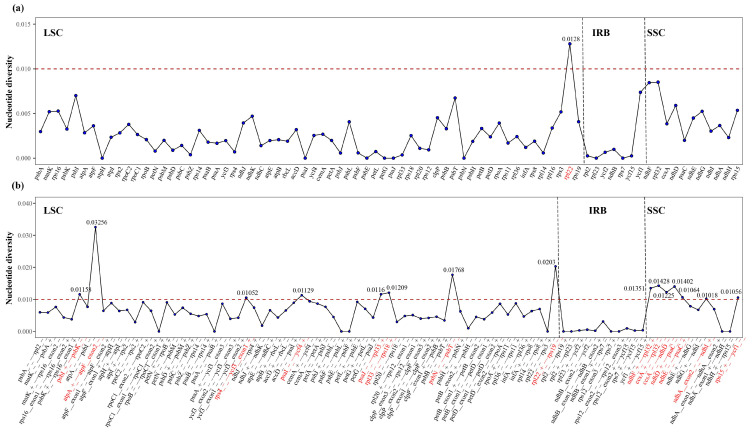
(**a**) Nucleotide polymorphism (Pi) analysis based on 80 CDS genes. (**b**) Nucleotide polymorphism (Pi) analysis based on the intergenic regions between the 80 CDS genes. Genes and intergenic regions are arranged counterclockwise, starting from the junction between SSC and IRa. The red dashed line represents a value set at Pi = 0.010, and the bold red text indicates genes with Pi values greater than 0.01.

**Figure 6 plants-14-00649-f006:**
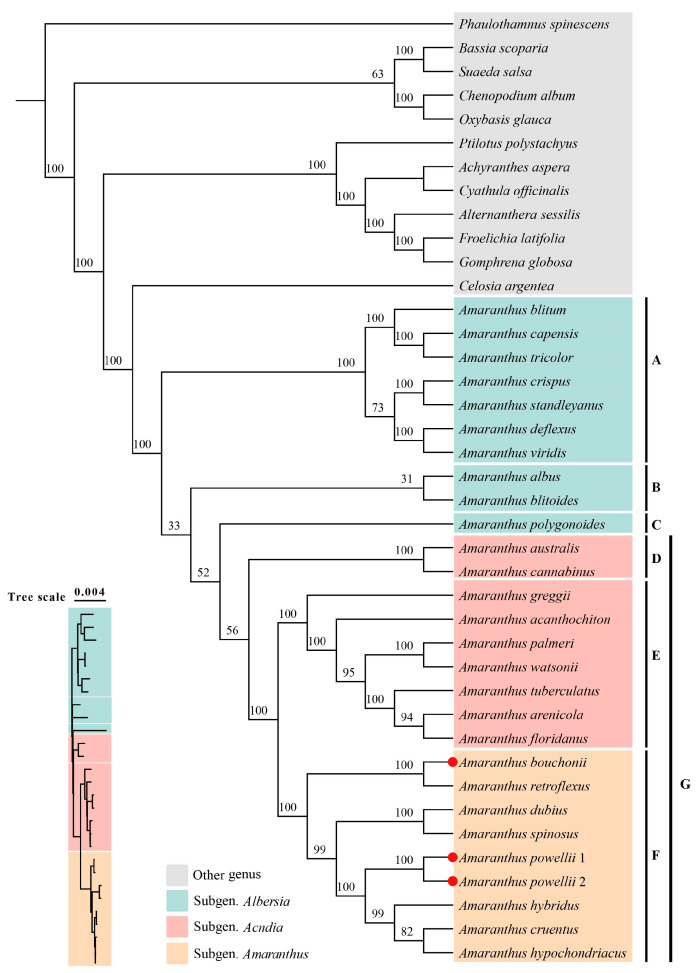
Maximum likelihood tree of *Amaranthus* based on 80 CDS sequences. The gray shaded blocks represent outgroups, the light green shaded blocks represent the subgenus *Albersia*, the light red shaded blocks represent the subgenus *Acnida*, and the light orange shaded blocks represent the subgenus *Amaranthus*. Red circles highlight the two species reported for the first time in this study. The letters A–G denote distinct phylogenetic clades within *Amaranthus*.

## Data Availability

The chloroplast genome sequences for the species studied are publicly available at the National Center for Biotechnology Information (NCBI) under the following accession numbers: *A. powellii*_1 (PQ385849), *A. powellii*_2 (PQ385850), and *A. bouchonii* (PQ375111).

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
