# Peer review of "Comprehensive Chloroplast Genomic Insights into Amaranthus: Resolving the Phylogenetic and Taxonomic Status of A. powellii and A. bouchonii"

_plants, 2025, doi:10.3390/plants14050649_

Round 1
Reviewer 1 Report
Comments and Suggestions for Authors
The submitted MS contains new data on the structure of chloroplast genomes of two Amaranthus species and a phylogenetic analysis of the genus based on all currently known sequences of Amaranthus chloroplast genomes. The study is important as the next step in reconstruction the phylogeny of specific angiosperm groups and as it contains a critical evaluation of available data on the subject. There are no comments on the essence of the paper.
Author Response
Comments 1: The submitted MS contains new data on the structure of chloroplast genomes of two Amaranthus species and a phylogenetic analysis of the genus based on all currently known sequences of Amaranthus chloroplast genomes. The study is important as the next step in reconstruction the phylogeny of specific angiosperm groups and as it contains a critical evaluation of available data on the subject. There are no comments on the essence of the paper.
Respons: We appreciate your positive feedback on our work.
Reviewer 2 Report
Comments and Suggestions for Authors
The manuscript with ID plants-3344668 titled: “Chloroplast Genomic Perspectives on the Molecular Evolution and Phylogenetic Relationships of Amaranths (Amaranthus , Amaranthaceae)” by Han, Yan, and Qi reports results of analyses on chloroplast genomes from Amarantus. Authors focused on resolving the phylogenetic and taxonomic status of two species of the genus, A. bouchonii and A. powellii. To achieve the aims of their study, authors obtained sequences of complete chloroplast genomes of these species and used their data in phylogenetic analyses of 80 protein-coding genes of chloroplast genome in a sample of 27 species of Amaranthus. These analyses confirmed monophyly of the genus and resolved phylogenetic relationships of A. bouchonii and A. powellii, supporting the view that these species are not closely related. In addition, authors reported results of comparative structural and selection pressure analyses of complete chloroplast genomes from their sample. These analyses respectively indicated highly conserved structure, gene content, and organization across the chloroplast genomes in the genus, and predominantly purifying selection across genes. Finally, authors identified 16 relatively highly variable regions that can serve as valuable molecular markers for species identification within Amaranthus. The study is well written and provides some useful insights into phylogenetic relationships within the genus. I intend to recommend this manuscript for publication in Plants, provided the authors will respond properly to my major concerns as regards this study.
Major comments
I found that the references in Introduction are not properly reported. Ref. 1 seems to be misplaced, making all references up to reference number 22 shifted. Please move this reference to its proper place (likely among numbers 22-27) and re-number all other references from 1 to 21.
I also think that Tables 1 and 2 do not represent particularly interesting results. The information summarized in the tables either reflects minor variation between species of Amaranthus in chloroplast genome length and GC content (Table 1) or repeats well-known from multiple earlier studies composition of functional genes of the chloroplast genome of Angiosperms (Table 2). The figure illustrates a graphical representation of the structure of chloroplast genome. These tables should be moved to Supplementary Information (SI). Authors should not distract the attention of the readers to a well-known information at the very beginning of reporting results of their study. Instead, they should focus on reporting the results, which are original and contribute new data to the field.I suggest moving the whole content of Chapter 2.1 to SI.
Minor comments
Introduction
L 53-56. A reference for the USDA database is required here.
L 63 “been debated” was repeated twice. Please correct.
L. 64. Reference 20 is very unlikely to be correctly cited. Please correct.
L. 76-79. “…insufficient sample sizes, unclear species identification, and low-resolution data…”
These issues are generally attributed to molecular studies. However, you write “…these classifications may not accurately reflect the evolutionary history of the genus.” Under “these classifications” I understand the classical taxonomy of the genus. Please clarify and specify what issues are related to molecular analyses, and what are characteristic of classical taxonomy.
L 89. Please spell out “SRA”. This abbreviation may not be familiar to most of the readers.
Results
L 191-192. The first sentence belongs to Material and Methods, not to Results. Please move it to M&M.
L 212-213. The same as for Chapter 2.5. The first sentence does not belong here.
L 234-235. Each clade is monophyletic (by definition). Therefore, a clade cannot “exhibit a paraphyletic relationship” with another clade. Clade D is sister to the rest of Clade G, but this topology received a very weak BS. Please correct as suggested.
L 229-231 The description of the results of phylogenetic analyses is often poor. Clade B is not supported (BS<0.5), so it should not be reported at all. Given the very low BS for relationships between Clades A, B, C, and D, these relationships should be interpreted as a polytomy. In the reported tree, only relationships between Clades E and F are strongly supported and should be mentioned. The chapter should be completely re-written. Only the part of this chapter on lines 235-242 can accepted.
As an example of the proper style, I suggest the following:
“Specifically, Clade A, comprising seven species (A. blitum, A. capensis, A. tricolor, A. crispus, A. standleyanus, A. deflexus, and A. viridis), received strong support (bootstrap support [BS] = 100) and together with A. albus, A. blitoides, A. polygonoides, and Clades D (two species of subgen. Acnidia) and G it forms a poorly resolved polytomy. Within Clade G, Clade E (the rest of subgen. Acnidia) and Clade F (subgen. Amaranthus) are sister to each other with strong statistical support (BS = 100).”
L 266 Grammatically incorrect.
L. 270 Point instead of comma.
L 277. Please delete the extra comma.
L 367. Clades are monophyletic by definition. “Monophyletic clades” is a tautology.
Author Response
Comments 1: The manuscript with ID plants-3344668 titled: “Chloroplast Genomic Perspectives on the Molecular Evolution and Phylogenetic Relationships of Amaranths (Amaranthus , Amaranthaceae)” by Han, Yan, and Qi reports results of analyses on chloroplast genomes from Amarantus. Authors focused on resolving the phylogenetic and taxonomic status of two species of the genus, A. bouchonii and A. powellii. To achieve the aims of their study, authors obtained sequences of complete chloroplast genomes of these species and used their data in phylogenetic analyses of 80 protein-coding genes of chloroplast genome in a sample of 27 species of Amaranthus. These analyses confirmed monophyly of the genus and resolved phylogenetic relationships of A. bouchonii and A. powellii, supporting the view that these species are not closely related. In addition, authors reported results of comparative structural and selection pressure analyses of complete chloroplast genomes from their sample. These analyses respectively indicated highly conserved structure, gene content, and organization across the chloroplast genomes in the genus, and predominantly purifying selection across genes. Finally, authors identified 16 relatively highly variable regions that can serve as valuable molecular markers for species identification within Amaranthus. The study is well written and provides some useful insights into phylogenetic relationships within the genus. I intend to recommend this manuscript for publication in Plants, provided the authors will respond properly to my major concerns as regards this study
Response: We sincerely appreciate you thoughtful comments and positive assessment of our study. We are pleased that the reviewer finds our work valuable in elucidating the phylogenetic relationships within Amaranthus. We will carefully address the major concerns raised and revise the manuscript accordingly to enhance its clarity and overall quality.
Comments 2: I found that the references in Introduction are not properly reported. Ref. 1 seems to be misplaced, making all references up to reference number 22 shifted. Please move this reference to its proper place (likely among numbers 22-27) and re-number all other references from 1 to 21.
Response: We appreciate you careful attention to the reference order. We have thoroughly reviewed the introduction and relocated reference 1 ("Comparative Analysis of Phylogenetic Relationships of Grain Amaranths and Their Wild Relatives") to the section discussing phylogenetic analysis. All references have been renumbered accordingly. Additionally, we have incorporated relevant literature on the origin and diversity of Amaranthus species to provide a more comprehensive background. We hope these revisions improve the clarity and completeness of the introduction.
Comments 3: I also think that Tables 1 and 2 do not represent particularly interesting results. The information summarized in the tables either reflects minor variation between species of Amaranthus in chloroplast genome length and GC content (Table 1) or repeats well- known from multiple earlier studies composition of functional genes of the chloroplast genome of Angiosperms (Table 2). The figure illustrates a graphical representation of the structure of chloroplast genome. These tables should be moved to Supplementary Information (SI). Authors should not distract the attention of the readers to a well-known information at the very beginning of reporting results of their study. Instead, they should focus on reporting the results, which are original and contribute new data to the field. I suggest moving the whole content of Chapter 2.1 to SI.
Response: We sincerely appreciate you suggestion regarding Tables 1 and 2 and the content in Section 2.1. We agree that some of the general chloroplast genome features, such as the overall genome size and functional gene composition, are well-documented in previous studies and may not require extensive emphasis in the main text. However, we believe that a concise description of the basic structural characteristics of Amaranthus chloroplast genomes is necessary to establish context before presenting our novel findings.
To address this, we have significantly condensed Section 2.1, retaining only the key genome characteristics that are directly relevant to the comparative analyses that follow. The detailed breakdown of gene categories and length variations, along with Tables 1 and 2, has been moved to the Supplementary Information as suggested. This revision allows us to maintain a logical flow while ensuring that the main text focuses primarily on novel contributions, avoiding unnecessary distraction for readers. We hope this approach balances the need for background information while adhering to you recommendation.
Minor comments Introduction
L 53-56. A reference for the USDA database is required here.
Response: We have now included the appropriate reference for the USDA database in L 52–54 to ensure proper citation and support for the information presented.
L 63 “been debated” was repeated twice. Please correct.
Response: Done.
L 64. Reference 20 is very unlikely to be correctly cited. Please correct.
Response: Thank you for pointing out the issue with reference 20. We have removed the incorrect citation and made the necessary corrections.
L 76-79. “…insufficient sample sizes, unclear species identification, and low-resolution data…”
These issues are generally attributed to molecular studies. However, you write “…these classifications may not accurately reflect the evolutionary history of the genus.” Under “these classifications” I understand the classical taxonomy of the genus. Please clarify and specify what issues are related to molecular analyses, and what are characteristic of classical taxonomy.
Response: We appreciate you careful reading and valuable feedback regarding the distinction between issues related to molecular analyses and classical taxonomy. To improve clarity, we have revised the sentence as follows:
"Additionally, some molecular studies have faced challenges, including insufficient sample sizes, unclear species identification, and the use of low-resolution genetic markers, leading to conflicting phylogenetic inferences."
L 89. Please spell out “SRA”. This abbreviation may not be familiar to most of the readers.
Response: Thank you for your suggestion. We have now spelled out "SRA" as "Sequence Read Archive (SRA)" in L 105 to ensure clarity for all readers.
Results
L 191-192. The first sentence belongs to Material and Methods, not to Results. Please move it to M&M.
Response: Done.
L 212-213. The same as for Chapter 2.5. The first sentence does not belong here.
Response: Done.
L 234-235. Each clade is monophyletic (by definition). Therefore, a clade cannot “exhibit a paraphyletic relationship” with another clade. Clade D is sister to the rest of Clade G, but this topology received a very weak BS. Please correct as suggested.
Response: We sincerely appreciate you insightful comment regarding the terminology. In response, we have revised the description to ensure accuracy in reflecting the phylogenetic relationships. Specifically, we have corrected the phrasing to avoid the incorrect use of "paraphyletic relationship" and now explicitly state that Clade D is the first diverging lineage within Clade G and is sister to the rest of Clade G (Clades E + F). Additionally, we have clarified the relationships among Clades E and F to align with the suggested correction.
L 229-231 The description of the results of phylogenetic analyses is often poor. Clade B is not supported (BS<0.5), so it should not be reported at all. Given the very low BS for relationships between Clades A, B, C, and D, these relationships should be interpreted as a polytomy. In the reported tree, only relationships between Clades E and F are strongly supported and should be mentioned. The chapter should be completely re-written. Only the part of this chapter on lines 235-242 can accepted.
As an example of the proper style, I suggest the following:
“Specifically, Clade A, comprising seven species (A. blitum, A. capensis, A. tricolor, A. crispus, A. standleyanus, A. deflexus, and A. viridis), received strong support (bootstrap support [BS] = 100) and together with A. albus, A. blitoides, A. polygonoides, and Clades D (two species of subgen. Acnidia) and G it forms a poorly resolved polytomy. Within Clade G, Clade E (the rest of subgen. Acnidia) and Clade F (subgen. Amaranthus) are sister to each other with strong statistical support (BS = 100).”
Response: We appreciate you insightful feedback regarding the clarity of the phylogenetic results in lines 238-245. As suggested, we have removed Clade B from the results due to its low bootstrap support (BS). Additionally, the relationships among Clades A, C, and D, which received weak support, have been interpreted as a polytomy. To enhance clarity, we have rewritten this section, emphasizing the strongly supported relationship between Clades E and F. We have also used the suggested example as a reference to ensure precise and accurate reporting of the phylogenetic relationships.These revisions have been incorporated into the updated manuscript.
L. 266 Grammatically incorrect.
Response: Done.
L. 270 Point instead of comma.
Response: We appreciate your suggestion. The comma has been replaced with a point as you advised.
L 277. Please delete the extra comma.
Response: Thank you for your attention to detail. We have removed it as requested.
L 367. Clades are monophyletic by definition. “Monophyletic clades” is a tautology.
Response: We understand your point regarding the tautology. The phrase "monophyletic clades" has been revised to simply "clades" to avoid redundancy.

Reviewer 3 Report
Comments and Suggestions for Authors
Reviewer 1
The article “Comprehensive Chloroplast Genomic Insights into Amaran-2 thus: Resolving the Phylogenetic and Taxonomic Status of A. powellii and A. bouchonii, authored by Jizhe Han, Chuhang Lin, Tingting Zhu, Yonghui Liu, Jing Yan, Xiaoling Yan, and Zhechen Qi focused into two species of Amaranthus, A. bouchonii and A. powellii, primarily to resolve a taxonomic status of these two species by including in their analysis a total of 27 species of this genus.
I found that article described clearly and detailed the structural differences of the chloroplast genomes for these 27 species. The figures inserted in the text are of bad quality, however, the files attached in JPG and PDF format have excellent quality in order to check easily their content. The supplementary files were properly prepared and they were clearly described; moreover, the eight supplementary tables contained in the excel file have additional supporting information for the article.
However, I consider that the article did not establish the biological problem to resolve. I was confused because in the Abstract it seems that phylogenetic perspective was treated as equal to taxonomy. Later in the introduction (lines 58-71) authors presented a taxonomic problem, since one of this species was considered as a subspecies. However, the immediate paragraph begins with a phylogenetic proposal (line 72). It is well recognized that phylogeny has provided auxiliary results to resolve taxonomic problems. However, I consider that authors have to write some antecedents to support the use of phylogenies for taxonomic problems. In addition, the biological problem of this article needs to be formally written. Moreover, authors clearly have to address which types of phylogenetic results (a tree species, a gene tree?), or which of the abundant and detailed results related to the structural-molecular variation, may be hypothesized/expected to elucidate/resolve the questions of the article.
I added directly on the pdf of the article various general comments. Please check carefully the pdf attached to this list of comments. I wrote abundant comments along the whole article, and particularly in the section the discussion. I consider complete paragraphs were redacted as antecedents and these were not truly discussed. Then, as these paragraphs are written they should be placed in the introduction, or really write a discussion.
Introduction. I consider that some paragraphs included in the Discussion may be useful in the Introduction section in order to put in context the biological problem to be resolved.
Methods. I consider that methods used for experimental as well as for the bioinformatic processes were adequate. I suggest that authors include the size of matrix used for phylogenetic analysis, and the equation for Pi be included.
Discussion. In addition to the comments here written, check carefully my comments directly written on the PDF of the submitted manuscript.
General comments
In addition, I recommend that in the abstract authors present that the 27 studied species are grouped into # subgenera, and give the taxonomic contest: The family groups # taxa, Amaranthus # taxa, which are grouped into # subgenera. I recommend, that in such part of the abstract authors write the biological problem/issues that they want to resolve; and consequently link with the reasons that they propose that a detailed characterization of the structural and molecular variation will resolve such issues.
Authors referred various published articles that reported complete chloroplast genome for distinct taxa, and they included keystone references for the central theme focused on chloroplast genome. However, since the results presented in the submitted article did not differ strongly with those already published and carried out with Amaranthus taxon, is convenient recognize such similarities in size, gene composition, structural conserved arrangement, etc. For example, with other species, such as: Xu XY, Yan J, Li HR, Feng YQ, Qi ZC, Yan XL. The complete chloroplast genome sequence of spleen amaranth (Amaranthus dubius Mart. ex Thell., Amaranthaceae). Mitochondrial DNA B Resour. 2021 Oct 23;6(11):3267-3268. doi: 10.1080/23802359.2021.1992318. PMID: 34712806; PMCID: PMC8547846.

Author Response
Please see the PDF file.

Round 2
Reviewer 2 Report
Comments and Suggestions for Authors
Thank you for your careful responses to my comments. I believe the revision of your study can be accepted for publiocation in Plants.